# *Corallococcus soli* sp. Nov., a Soil Myxobacterium Isolated from Subtropical Climate, Chalus County, Iran, and Its Potential to Produce Secondary Metabolites

**DOI:** 10.3390/microorganisms10071262

**Published:** 2022-06-21

**Authors:** Zahra Khosravi Babadi, Ronald Garcia, Gholam Hossein Ebrahimipour, Chandra Risdian, Peter Kämpfer, Michael Jarek, Rolf Müller, Joachim Wink

**Affiliations:** 1Department of Microbiology and Microbial Biotechnology, Faculty of Life Sciences and Biotechnology, Shahid Beheshti University GC, Tehran 1983969411, Iran; g-ebrahimi@sbu.ac.ir; 2Microbial Strain Collection, Helmholtz Centre for Infection Research (HZI), Inhoffenstraße 7, 38124 Braunschweig, Germany; chandra.risdian@helmholtz-hzi.de; 3Helmholtz Institute for Pharmaceutical Research Saarland (HIPS), Helmholtz Centre for Infection Research (HZI), Saarland University, Campus E8 1, 66123 Saarbrücken, Germany; ronald.garcia@helmholtz-hips.de (R.G.); rolf.mueller@helmholtz-hips.de (R.M.); 4German Centre for Infection Research (DZIF), Partner Site Hannover-Braunschweig, 38124 Braunschweig, Germany; 5Research Unit for Clean Technology, National Research and Innovation Agency (BRIN), Bandung 40135, Indonesia; 6Department of Applied Microbiology, Justus Liebig University Gießen, 35392 Gießen, Germany; peter.kaempfer@umwelt.uni-giessen.de; 7Genome Analytics, Helmholtz Centre for Infection Research (HZI), Inhoffenstraße 7, 38124 Braunschweig, Germany; michael.jarek@helmholtz-hzi.de

**Keywords:** myxobacteria, myxococcales, *Corallococcus*, *Corallococcus soli*

## Abstract

A novel myxobacterial strain ZKHCc1 1396^T^ was isolated in 2017 from a soil sample collected along Chalus Road connecting Tehran and Mazandaran, Iran. It was a Gram-negative, rod-shaped bacterial strain that displayed the general features of *Corallococcus,* including gliding and fruiting body formation on agar and microbial lytic activity. Strain ZKHCc1 1396^T^ was characterized as an aerobic, mesophilic, and chemoheterotrophic bacterium resistant to many antibiotics. The major cellular fatty acids were branched-chain *iso*-C_17:0_ 2-OH, *iso*-C_15:0_, *iso*-C_17:1_, and *iso*-C_17:0_. The strain showed the highest 16S rRNA gene sequence similarity to *Corallococcus*
*terminator* CA054A^T^ (99.67%) and *C. praedator* CA031B^T^ (99.17%), and formed a novel branch both in the 16S rRNA gene sequence and phylogenomic tree. The genome size was 9,437,609 bp, with a DNA G + C content of 69.8 mol%. The strain had an average nucleotide identity (ANI) value lower than the species cut-off (95%), and with the digital DNA–DNA hybridization (dDDH) below the 70% threshold compared to the closest type strains. Secondary metabolite and biosynthetic gene cluster analyses revealed the strain’s potential to produce novel compounds. Based on polyphasic taxonomic characterization, we propose that strain ZKHCc1 1396^T^ represents a novel species, *Corallococcus soli* sp. nov. (NCCB 100659^T^ = CIP 111634^T^).

## 1. Introduction

Myxobacteria are Gram-negative, rod-shaped bacteria belonging to the phylum *Myxococcota* [1] and are considered unique for their social behavior and complex developmental growth stages. In many myxobacteria, nutrient-limiting conditions enable the vegetative cells to swarm, aggregate, and form multicellular fruiting bodies [2,3,4,5]. Within the fruiting bodies, resistant and dormant myxospores are contained to ensure the next generation of cells. Myxobacteria are widely and commonly distributed in nature, including in topsoil, animal dung, decaying plants, bark of trees [2,5], and even in halophilic environments [6,7,8,9]. Isolation of myxobacteria in the past decades was mainly driven by natural product application, including the discovery of new antibiotics, and anticancer and antiviral compounds [10,11,12,13].

Myxobacteria form a relatively homogeneous cluster based on 16S rRNA gene-based phylogenetic analysis [14,15,16]. In the last decade, many new genera and families of myxobacteria with unprecedented characteristics were discovered [17,18,19,20,21]. The discovery of several new *Corallococcus* species in the previous year [22] was not a surprise, since they seem common and widespread in the environment. In the ten described *Corallococcus* species with validated names, eight were primarily taxonomically described based on the draft genomic data [22]. The genus *Corallococcus* is known for its rippling swarm; hard, non-sporangiole-type fruiting body; and rounded myxospores [23], which makes morphology an important component in strain characterization. These combined growth features help distinguish this group of bacteria at the very early stage of strain isolation.

*Corallococcus* appears to gain relatively more attention due to the bioactive compounds discovered in this genus. The strains of *C. coralloides* had been described for several compounds, including corallopyronins A, B, and C [24], corallorazine A [25], and coralmycins A and B [26], which are known to have antibacterial properties. Of these antibiotics, only the corallopyronin BGC information from *Corallococcus coralloides* strain B035 [27] is available in the MIBiG database (https://mibig.secondarymetabolites.org/, accessed on 15 December 2021).

The present study describes a novel species of myxobacteria in the genus *Corallococcus*, and demonstrates its potential to produce novel secondary metabolites.

## 2. Materials and Methods

### 2.1. Isolation and Maintenance

Strain ZKHCc1 1396^T^ was isolated at the Helmholtz Centre for Infection Research (HZI) in autumn 2017 from a soil sample collected in October 2015 along the Chalus Road between Tehran and Mazandaran 36°39′18.00″ N, 51°25′13.44″ E, Iran. Chalus Road is a mountainous area connecting Tehran to several northern cities. Its climate is classified as temperate and warm, with more rain in the winter (www.piniran.com, accessed on 4 March 2019). The average annual temperature is 15.7 °C, while the average yearly rainfall is 1081 mm. Its highest temperature (ca. 25.6 °C) peaks in August, while its lowest (ca. 7.2 °C) occurs in February.

The isolation of myxobacteria was based on the standard bacterial baiting technique [3,4,5]. The soil sample was poured to water agar with streaks of living *E. coli* K-12 DSM 498 bait. The strain was purified by repeated transfers of the swarm edge onto new mineral salt agar [3,4,5]. Axenic culture was maintained on standard VY/2 agar [3,4] and CY–H medium (*w/v*: 50% CY medium (0.3% Casitone (Difco, Franklin Lakes, NJ, USA), 0.1% yeast extract (Difco), 0.1% CaCl_2_·2H_2_O); 50% H medium (0.2% soy meal flour (Hensel, Magstadt, Germany), 0.2% glucose (Sigma-Aldrich, St. Louis, MO, USA), 0.8% soluble starch (Carl Roth, Karlsruhe, Germany), 0.2% yeast extract (Difco), 0.1% CaCl_2_, 0.1% MgSO_4_), 50 mM HEPES, and 8 mg Fe-EDTA; adjusted to pH 7.4 with KOH before autoclaving), supplemented with 500 µg ml^−1^ vitamin B_12_. Culture in CY–H broth was rotary shaken at 160 r.p.m. for seven days. Both the VY/2 agar plate culture and CY–H broth were incubated at 30 °C. Sample for long-term storage in a −80 °C freezer were prepared from an actively growing CY–H culture and preserved using 20–25% (*v/v*) glycerol as a cryoprotectant [4].

### 2.2. Physiology and Chemotaxonomy

Growth morphology characterization was performed on standard nutrient-lean media, including VY/2 and water agar baited with *E. coli* K-12, and in standard Casitone-containing CY [23] and CY–H media. All solid media in this study contained 1.6% (*w/v*) Bacto agar. Fruiting bodies and myxospores were observed from previously described nutrient-lean agar media, while the vegetative cells were studied after cultivation in CY–H broth after six days of shaking (160 r.p.m., 30 °C). Swarming was observed on lean media and as well as on Casitone-containing agars. The fruiting bodies and swarm colonies were examined using an Olympus SZX12 stereomicroscope, while the vegetative cells and myxospores were studied using a Zeiss AX10 phase-contrast microscope, photographed using a Zeiss Axiocam MRC camera, and analyzed using AxioVision LE software.

Gram-staining, oxidase, and catalase tests were based on previously described methods [18,28]. The API ZYM^®^ (bioMérieux) and API^®^ Coryne reactions were conducted following the manufacturer’s instructions. Temperature tolerance of the novel strain was determined at 18, 25, 30, 35, 37, and 40 °C, while pH tolerance was tested at pH 5.0–9.0 with intervals of pH 0.5. Both temperature and pH determination were performed in VY/2 agar and were assessed based on the colony growth.

Antibiotic resistance of the novel isolate was tested on VY/2 agar with 50 µg ml^−1^ antibiotic concentration. The tested antibiotics were ampicillin, amikacin, cefotaxime (Carl Roth), ceftazidime, imipenem, gentamicin, and trimethoprim-sulfamethoxazole (Sigma-Aldrich). All antibiotics were filter-sterilized before being added to the autoclaved agar, which was cooled down to 55 °C before plating.

Microbial predation of the novel myxobacterium was tested using *Bacillus subtilis* DSM 10^T^, *Micrococcus luteus* DSM 1790, *Escherichia coli* DSM 1116, and *Wickerhamomyces anomalus* DSM 6766^T^. Predation was evaluated for clearing of the baited strain, which indicated cell lysis. Degradation of cellulose and chitin was determined based on previously described methods [17], while agar degradation was determined in all solid media, using Bacto agar (1.6% *w/v*) as a solidifying agent.

The fatty acid extraction was performed using the fatty acid methyl ester method (FAME) [29,30]. The strain was cultivated in 50 mL myxovirescin medium (*w/v*: 1% soy peptone, 0.025% MgSO_4_, 0.005% CaCl_2_, 1 mg/L CoCl_2_, 100 mM HEPES; adjusted to pH 7.0 with KOH before autoclaving) under shaking conditions (160 r.p.m., 30 °C, six days) before it was harvested by centrifugation (21,000× *g*, 10 min, 4 °C) and extracted for fatty acids. Analysis and identification of fatty acids was performed by GC–MS based on the standard method for myxobacteria [30].

### 2.3. Genome and Phylogenetic Analysis

For genomic DNA isolation, the cells were obtained from an actively growing CY–H culture and the DNA was extracted following the standard method for Gram-negative bacteria using the Puregene Core Kit A from Qiagen. The amplification of the 16S rRNA gene was performed using the universal primers F27 (5′-GAGTTTGATCCTGGCTCAGGA-3′) and R1525 (5′-AAGGAGGTGATCCAGCCGCA-3′) [17]. The amplified PCR products were purified using a Macherey Nagel NucleoSpin Kit, separated by gel electrophoresis (0.8% (*w/v*) agarose, at 70 V, for 45 min), and subsequently sequenced using primers F27 [18], R1525, R518, F1100, and R1100 [19]. The 16S rRNA gene sequence was aligned using the Cap contig assembly of the BioEdit Sequence Alignment Editor software version 7.0.5 [31].

The 16S rRNA gene sequence phylogenetic analysis was conducted using the GGDC web server (http://ggdc.dsmz.de/, accessed on 25 January 2022) [32]. Pairwise sequence similarities were calculated according to the method of Meier-Kolthoff et al. [33]. Phylogenies were inferred using the phylogenomics pipeline developed by DSMZ [34] adapted to single genes, and the sequence alignment was performed using MUSCLE [35]. Maximum likelihood (ML) and maximum parsimony (MP) trees were constructed using RAxML [36] and TNT [37], respectively. For ML, the autoMRE bootstrapping criterion [38], and a subsequent search for the best tree, was employed. In MP, 1000 replicates from bootstrapping were used in conjunction with tree bisection and reconnection branch swapping, and ten random sequence additional replicates. The sequences were evaluated for a compositional bias using the Χ^2^ test implemented in PAUP* [39].

The genome sequencing of strain ZKHCc1 1396^T^ was carried out using next-generation sequencing technology (Illumina) with MiSeq 600 cycle v3. De novo genome assembly was performed using a Unicycler [40]. Predicted genes, tRNA genes, rRNA genes, and other characteristics of the genome were annotated using PROKKA [41]. In addition, the annotated data from the Prokaryotic Genome Annotation Pipeline (PGAP) of NCBI [42] were also used for genomic comparisons of all *Corallococcus* type strain genomes. The possible contamination of the genomic data was evaluated using the ContEst16S algorithm to analyze the 16S rRNA gene fragments (https://www.ezbiocloud.net/tools/contest16s, accessed on 10 March 2020) [43]. The complete 16S rRNA gene sequence of the novel strain was extracted from its genome, and this was used for the phylogenetic analysis and percentage similarity comparisons with the closest type strains. The percentage DNA G + C content was determined based on the strain’s genome sequence.

The genomic sequence data of strain ZKHCc1 1396^T^ was uploaded in the Type Strain Genome Server (TYGS) (https://tygs.dsmz.de, accessed on 25 January 2022) for a whole-genome-based taxonomic analysis [44] with the recently introduced methodological updates and features [32]. Information on nomenclature, synonymy, and associated taxonomic literature was provided by the List of Prokaryotic names with Standing Nomenclature (LPSN, available at https://lpsn.dsmz.de, accessed on 25 January 2022) [45].

The uploaded genome was compared against all type strain genomes in the TYGS database using the MASH algorithm [46]. Additionally, the genome data of *Corallococcus silvisoli* c25j21^T^ (JAAAPJ000000000) was also added separately from the NCBI website (https://www.ncbi.nlm.nih.gov/, accessed on 26 January 2022) because it was not yet listed in the TYGS database. The closest type strains were chosen based on the smallest MASH distances and the 16S rRNA gene sequences. Extraction of 16S rRNA gene sequences from the genome was completed using RNAmmer [47], and was subsequently BLASTed [48] against the type strain’s 16S rRNA gene sequences available in the TYGS database. This was used as a proxy to search for the top 50 matching type strains (according to the bitscore) and to calculate the distances using the Genome BLAST Distance Phylogeny approach (GBDP) under the “coverage” and distance formula d5 algorithm [49]. These distances were used in determining the closest type strain genomes.

Phylogenomic inference was performed using the GBDP. Intergenomic distances were inferred using “trimming” and distance formula d5 algorithm [49] with 100 distance replicates. The digital DDH values (dDDH) and confidence intervals were calculated using the GGDC 3.0 recommended settings [46,49]. The resulting intergenomic distances were used to infer a balanced minimum evolution tree with FASTME 2.1.6.1 branch support, and include SPR post-processing [50]. Branch support was calculated from 100 pseudo-bootstrap replicates each. The phylogenetic trees were rooted at the midpoint [51], and was visualized using a PhyD3 program [52].

The genome sequences of the 11 closely related type strains were retrieved from the NCBI database (https://www.ncbi.nlm.nih.gov/, accessed on 26 January 2022). Average nucleotide identity (ANI) was calculated using the algorithm OrthoANIu (OrthoANI using USEARCH) [53], while the digital DNA–DNA hybridization (dDDH) was determined using the Genome-to-Genome Distance Calculator (GGDC) version 2.1, http://ggdc.dsmz.de (accessed on 26 January 2022) [49].

#### BiG-SCAPE Analysis

Genome data for all of the type strains of *Corallococcus* species were downloaded from the NCBI database. The genome data were analyzed using AntiSMASH version 6.0.0 (available at https://antismash.secondarymetabolites.org/, accessed on 2 February 2022) to identify the secondary metabolite gene clusters using the “relaxed” strictness setting [54,55]. All of the predicted biosynthetic gene clusters (BGCs) were then analyzed using the BiG-SCAPE program (version 1.1.2 (3 June 2021)), with the MiBIG database (version 2.1) as a reference [56,57], and Pfam database version 34.0 [58]. Some parameters that were used include a distance cut-off score of 0.3, and the search terms “hybrid” and “mix”. Generated networks were visualized with Cytoscape (version 3.8.2) [59].

### 2.4. Extract Production, Antimicrobial Assay, and Extract Analysis

The strain ZKHCc1 1396^T^ pre-culture was grown in a 100 mL flask containing 20 mL CY–H broth and incubated on a rotary shaker (160 r.p.m.) for 7–14 days at 30 °C. To screen for secondary metabolites, the resultant cultures (20 mL) were transferred in 250 mL flasks containing 100 mL of the production media, including E medium, CY medium, P medium, POL medium, S medium, M medium, and myxovirescin medium (Appendix A), and supplemented with 2% (*v/v*) XAD-16 adsorber resin. After 14 days of incubation at 30 °C, the resins and cells were collected together by filtering through a fine metal mesh. They were extracted with 70 mL acetone for one hour, filtered with filter paper, and concentrated in vacuo at 40 °C using a rotary evaporator (Heidolph, Schwabach, Germany). The dried extract was resolved in 1 mL methanol and was used (20 µL/strain) in the antimicrobial assay against *E. coli* DSM 1116, *E. coli* TolC, *S. aureus* Newman, *C*. *albicans* DSM 1665, *Pseudomonas aeruginosa* DSM 19882, *B*. *subtilis* DSM 10^T^, *Micrococcus luteus* DSM 1790, *M*. *smegmatis* ATCC 700084, *Chromobacterium violaceum* DSM 30191^T^, *M*. *hiemalis* DSM 2656^T^, and *Wickerhamomyces anomalus* DSM 6766^T^. These assay strains were obtained from the Microbial Strain Collection Group (MISG) of Helmholtz Centre for Infection Research (HZI) in Braunschweig, Germany. These bacteria were cultivated in Mueller–Hinton broth (Merck) to obtain OD_600_ of 0.01. Yeast was cultivated in Mycosel broth [60] to obtain OD_600_ of 0.05.

The crude extract, which showed high and interesting biological activity, was chosen for further analysis using HPLC-DAD Agilent 1260 series coupled with a MaXis ESI–TOF mass spectrometer (Bruker Daltonics, Bremen, Germany). Column C18 Acquity UPLC BEH (Ultra Performance Liquid Chromatography Ethylene-Bridged Hybrid, Waters) and two solvents (solvent A: H_2_O + 0.1% formic acid; solvent B: ACN + 0.1% formic acid) were used in the HPLC system. The compound separation was performed with a flow rate of 0.6 mL/min, column temperature of 40 °C, and the gradient condition was as follows: 5% B in 0–0.5 min, 5–100% B in 0.5–20 min, and 100% B in 20–25 min [61,62]. Chromatogram and spectrum analysis was conducted using Compass DataAnalysis Version 4.4 (Bruker Daltonics, Billerica, MA, USA). Fractions were selected based on retention time every two minutes in the range of 1.8–20 min from the base peak chromatogram (BPC). For compound prediction, the detected accurate mass (with ±0.01 Da) was manually searched for using the database of Dictionary of Natural Products version 30.1.

## 3. Results and Discussion

### 3.1. Taxonomic Identification

Strain ZKHCc1 1396^T^ showed the characteristic features of myxobacteria, which include swarming, fruiting, and myxospore formation (Figure 1). In all solid agar media, the colony produced a coherent swarm after inoculation at the center of the Petri dish (Figure 1a,b). On VY/2 agar, the colony spread fast and formed dense orange fruiting bodies, while a halo was formed after clearing the Baker’s yeast cells (Figure 1a). In contrast, the growth on CY agar was slower and the colony appeared a darker orange due to swarming and some cell aggregates (Figure 1b). The strain produced a thin and transparent swarm with ripples and a flare-shaped pattern at the colony edges in VY/2 agar (Figure 1c), while veins were commonly produced in Casitone-containing CY agar (Figure 1d). Swarms remained on the surface of the agar with no diffusing pigment. Yellowish-to-orange, hard, coral- or horn-shaped fruiting bodies were observed in standard, nutrient-lean VY/2 and water agars, and measured 250–543.5 µm, commonly visible to the naked eye (Figure 1e,f). These unique fruiting body structures were not observed in CY and CY–H agars, but instead were replaced by orange cell aggregates. Fruiting bodies contained tightly packed, rounded, and optically refractile myxospores with a thick coat, and measured 1.3–2.2 µm in diameter (Figure 1g). In contrast, the vegetative cells were non-motile, phase-dark, flexuous, and nearly spindle-shaped rods that measured 4.0–7.6 µm (Figure 1h). All these growth stage characteristics fit within the genus *Corallococcus* [23].

Strain ZKHCc1 1396^T^ was catalase-positive, oxidase-negative, and stained Gram-negative. It showed positive API ZYM reactions (+3 to +5) to alkaline phosphatase, C8 esterase lipase, C14 lipase, trypsin, and acid phosphatase, and weak positive reactions (+1 to +2) to C4 esterase, leucine arylamidase, valine arylamidase, cysteine arylamidase, *α*-chymotrypsin, and naphthol-AS-BI-phosphohydrolase. API ZYM reactions were negative (0) to *α*- and *β*-galactosidase, *β*-glucuronidase, *α*- and *β*-glucosidase, *N*-acetyl-*β*-glucosaminidase, *α*-mannosidase, and *α*-fucosidase. In the API^®^ Coryne, only the gelatin hydrolysis and alkaline phosphatase exhibited positive reactions, which were both similar to those from the same tests using the API ZYM. All these differentiating characteristics with *Corallococcus* type strains are summarized in Table 1.

Strain ZKHCc1 1396^T^ exhibited colony growth at 18–35 °C but not at the higher temperatures tested (37 and 40 °C). The optimal growth of the strain was observed at 35 °C; this differs from most of the *Corallococcus* type strains except for AB050A^T^, which shows nearly the same pattern. The pH range of the isolate was determined between pH 5.5–10 and was optimal at pH 6.0–8.5, which is in the bracket for most *Corallococcus* type strains (Table 1).

Strain ZKHCc1 1396^T^ was susceptible to amikacin, cefotaxime, and ceftazidime, but not ampicillin, gentamicin, imipenem, or trimethoprim. The susceptibility to both cefotaxime and ceftazidime hallmarks the difference in the antibiotic profiles among *Corallococcus* (Table 1).

Cellulose powder, filter paper, chitin, and agar were not degraded or digested, suggesting that the new strain lacks cellulolytic, chitinolytic, and agarolytic activity, respectively. The lysis of the tested bacteria and yeast is not surprising, as it is a common characteristic of the genus *Corallococcus* and many other myxobacteria in the family *Myxococcaceae*.

The major cellular fatty acids of strain ZKHCc1 1396^T^ were *iso*-C_17:0_ 2-OH (31.0%), *iso*-C_15:0_ (15.8%), *iso*-C_17:1_ (11.7%), and *iso*-C_17:0_ (9.4%) (Table 2). The remarkably high amount of branched-chain fatty acids (94.2%) over the straight-chain type agrees with previous myxobacterial fatty acid studies on the genus *Corallococcus*, and thus differentiates it from the related genera *Myxococcus, Pyxidicoccus* (Garcia et al., 2011), and *Simulacricoccus* (Garcia et al. 2018). Among the branched-chain fatty acids, *iso*-C_15_:_0_, *iso*-C_16_:_0_, *iso*-C_17_:_0_, and *iso*-C_17:1_ were found to be the most abundant, accounting to 15.8%, 5.6%, 9.4%, and 11.7%, respectively. Moreover, *iso*-C_15:0_ was found in all *Corallococcus* type species (Appendix A), and seemed to be one of the major fatty acids in this genus, and as well as in the whole *Myxococcaceae* family [29]. The overall fatty acid patterns and their major types indicates that strain ZKHCc1 1396^T^ belongs to the genus *Corallococcus*, but differs from other type species in their fatty acid quantities.

The amplified, almost complete 16S rRNA gene sequence of strain ZKHCc1 1396^T^ was about 1501 bp, while the complete sequence obtained from the genome was 1536 bp, and was determined as identical. Based on the complete 16S rRNA gene sequence, the closest type strain similarities were *Corallococcus*
*terminator* CA054A^T^ (99.67%) and *C. praedator* CA031B^T^ (99.17%). Phylogenetic analyses revealed that strain ZKHCc1 1396^T^ clustered within the *Corallococcus* clade, but formed a separate branch with the closest type strains (Figure 2).

According to the PROKKA annotation, the assembled draft genome of strain ZKHCc1 1396^T^ (GenBank accession No. JAAIYO000000000) consisted of 9,437,609 bp and was characterized by 69.8 %mol G + C content. The genome was predicted to contain 7535 genes comprising 7453 protein-coding genes, 66 tRNA genes, three rRNA genes, and one copy each of the 5S rRNA, 16S rRNA, and 23S rRNA gene. No signs of contamination were found in the genome based on one copy of the 16S rRNA gene. In contrast, the number of genes, proteins, and RNAs varied in number based on the NCBI PGAP annotation pipeline. For comparison with the type strains, the PGAP annotation was used since all data are available in NCBI (Table 1). Strain ZKHCc1 1396^T^ differs among type strains of other *Corallococcus* type species by having the smallest genome size and the least number of genes, proteins, and tRNAs (Table 1).

The phylogenomic tree supports the novelty of strain ZKHCc1 1396^T^ as it forms a novel branch in the *Corallococcus* clade. The closest species type strain appears to be *Corallococcus praedator* CA031B^T^ and *C. terminator* CA054A^T^ (Figure 3). Furthermore, the difference of the isolated myxobacterium is indicated by the ANI and dDDH values (Table 3). All type strains compared have ANI values lower than the species cut-off (95%), and with dDDH scores below the 70% threshold value.

### 3.2. Comparison and Networking of the Secondary Metabolite Biosynthetic Gene Clusters (BGCs)

All of the eleven *Corallococcus* strains show a 100% similarity score for the geosmin gene cluster (Table 4), while nine strains exhibited 100% for the rhizomide A/rhizomide B/rhizomide C gene cluster. However, based on AntiSMASH evaluation of all the genomes of the *Corallococcus* type strains, none of them contained the corallopyronin BGC, including the type strain of *Corallococcus coralloides* DSM 2259^T^. High similarity scores (≥60%) were found in all strains for the BGC of VEPE/AEPE/TG-1, and nine strains for a carotenoid and myxochelin A/myxochelin B. Our results are in accordance with a previous study by Ahearne et al. [64], which suggested that all of the myxobacterial strains from the *Myxococcaceae* family contained the BGC of geosmin, VEPE/AEPE/TG-1, and a carotenoid. Strain ZKHCc1 1396^T^ appears to be closely similar in the BGC pattern of *Corallococcus terminator* CA054A^T^, but lacks the BGC for icosalide A/icosalide B.

In the nonribosomal peptide synthetase (NRPS) gene cluster, one BGC of strain ZKHCc1 1396^T^ formed a cluster with numerous edges with the BGCs of other *Corallococcus* type strains together with the BGC of VEPE/AEPE/TG-1 from the MIBiG (minimum information about a biosynthetic gene cluster) database (Figure 4). Two NRPS gene clusters of strain ZKHCc1 1396^T^ were found to have one edge. In the type I PKS (polyketide synthases) gene cluster, one BGC of strain ZKHCc1 1396^T^ had two edges, while in the other PKS gene cluster, it possessed three edges. Strain ZKHCc1 1396^T^ formed seven gene cluster families (GCFs) with the other strains in the NRPS–PKS hybrid gene clusters, whereas five GCFs were created in the RiPPs (ribosomally synthesized and post-translationally modified peptides) gene cluster that contained BGC of strain ZKHCc1 1396^T^. One GCF in the terpene gene cluster comprised one of the BGCs of strain ZKHCc1 1396^T^, which was connected with the some BGCs of other *Corallococcus* type strains, and a carotenoid gene cluster from MIBiG database. For the other BGCs, there were two GCFs containing the BGC of strain ZKHCc1 1396^T^. Overall, from the analysis of gene cluster network using the BiG-SCAPE platform, the BGCs of strain ZKHCc1 1396^T^ could form one or more GCFs to the other type strains of *Corallococcus* species in various types of BGCs.

From the various secondary metabolite production media, the cultivation in P medium was shown to have the most bioactive crude extract against Gram-positive bacteria (Figure 5). The BPC chromatogram of strain ZKHCc1 1396^T^ obtained from cultivation and extraction in P medium produced more than twenty high peaks (above the 90% relative intensity compared to the highest peak) in the range of 1.8–20 min. (Figure 6). The detected ion masses ranged from 209.1645 Da to 1371.9401 Da (Table 5), suggesting the presence of diverse compounds produced by strain ZKHCc1 1396^T^. Interestingly, no hit compound from DNP was found from the genus *Corallococcus*. Two hits were detected with similar masses to myxobacterial species *Chondromyces crocatus*. Analysis of fraction 19 showed no hits in the DNP database, suggesting the possibility of discovering a novel compound from strain ZKHCc1 1396^T^. Further study is needed to confirm the compounds produced by strain ZKHCc1 1396^T^, which can be conducted by the isolation and structure elucidation of the compounds.

### 3.3. Description of Corallococcus soli sp. Nov.

*Corallococcus soli* (so’li. L. gen. n. soli, of soil, referring to a myxobacterium isolated from a soil sample collected in Iran).

Vegetative cells are phase-dark and flexuous rods (4.0–7.6 µm) in CY–H medium, and move by gliding on agar. Swarm colonies are transparent in VY/2 and water agars, with a rippling pattern. Bright orange colonies with veins are evident in CY and CY–H agars. Fruiting bodies in VY/2 and water agars appear orange and very large (250–543.5 µm) forming cartilaginous, hard horns and coral-shaped structures. Myxospores packed in the fruiting bodies are refractile, spherical-to-ellipsoidal (1.3–2.2 µm diameter) with a distinct spore coat. Other characteristics include aerobic, mesophilic (18–35 °C, optimum at 35 °C), neutrophilic (pH 5.5–10, optimum at pH 6.0–8.5), Gram-negative, catalase-positive, and oxidase-negative. We found that agar, chitin, cellulose are not degraded by this strain. Moreover, the strain exhibits positive API ZYM reactions to alkaline phosphatase, C8 esterase lipase, C14 lipase, trypsin, and acid phosphatase, weak positive reactions to C4 esterase, leucine arylamidase, valine arylamidase, cysteine arylamidase, *α*-chymotrypsin, and naphthol-AS-BI-phosphohydrolase, and negative API ZYM reactions to *α*- and *β*-galactosidase, *β*-glucuronidase, *α*- and *β*-glucosidase, *N*-acetyl-*β*-glucosaminidase, *α*-mannosidase, and *α*-fucosidase. It can hydrolyze gelatin but unable to ferment glucose, ribose, xylose, mannitol, maltose, lactose, sucrose, or glycogen. Furthermore, it exhibits a negative API^®^ Coryne reaction to nitrate, pyrazinamidase, pyrrolidonyl arylamidase, and urease, and is resistant to ampicillin, gentamicin, imipenem, and trimethoprim, but sensitive to amikacin, cefotaxime, and ceftazidime. The major cellular fatty acids of this strain are iso-C_17:0_ 2-OH, iso-C_15:0_, iso-C_17:1_, and iso-C_17:0_. The DNA G + C content is 69.7 mol%, and the genome size of the type strain is 9,437,609 bp. The draft genome sequence is available under DDBJ/ENA/GenBank accession number JAAIYO000000000. The type strain is ZKHCc1 1396^T^ (=NCCB 100659^T^ = CIP 111634^T^), isolated in 2017 from a soil sample collected along Chalus Road between Tehran and Mazandaran in Iran

## 4. Conclusions

Based on polyphasic characterizations, which include morphological, physiochemical, and genomic studies, strain ZKHCc1 1396^T^ appears to represent a novel species of *Corallococcus*, for which we propose the name *Corallococcus soli* sp. nov. (type strain ZKHCc1 1396^T^ (=NCCB 100659^T^ = CIP 111634^T^)).

## Figures and Tables

**Figure 1 microorganisms-10-01262-f001:**
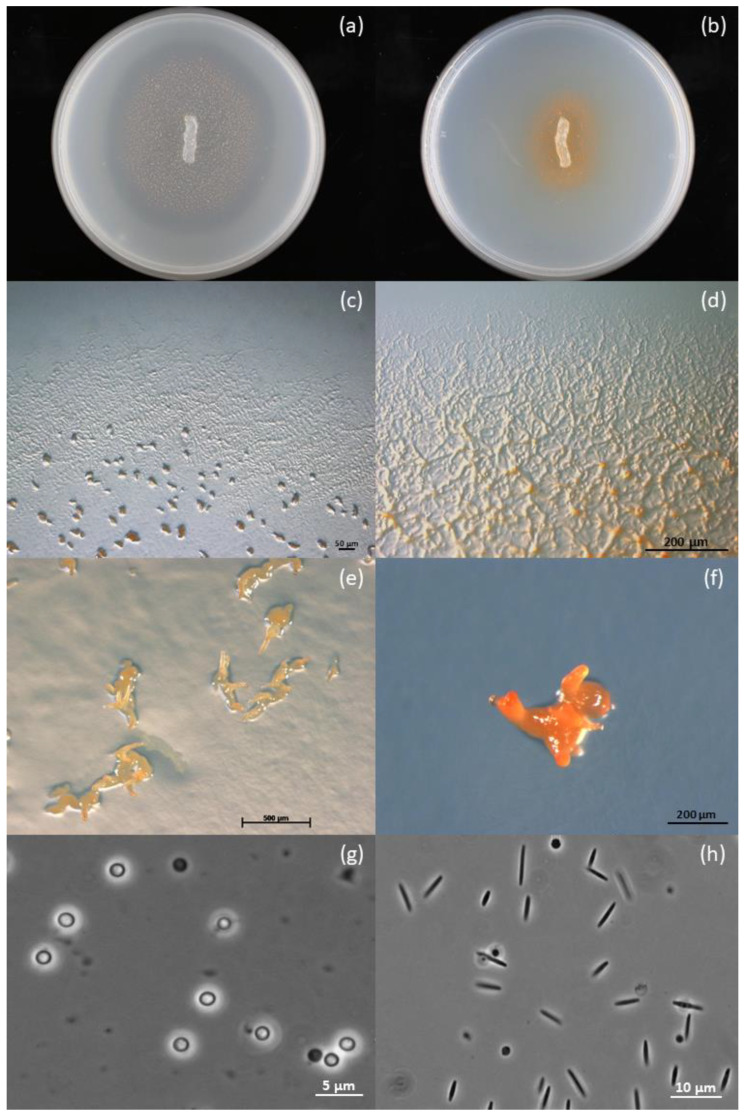
Growth morphologies of strain ZKHCc1 1396^T^. (**a**) Colony on VY/2 agar showing dense orange fruiting bodies around the agar inoculum and lysis of Baker’s yeast cells, as indicated by a halo around it. (**b**) Colony on CY agar showing darker orange color produced by the swarming cells. (**c**) Thin and transparent swarm on VY/2 agar with characteristic ripples and flares along the colony edges and fruiting bodies. (**d**) Swarm on CY agar with pronounced veins and some cell mounds. (**e**) Coral- or horn-shaped fruiting bodies produced on VY/2 agar. (**f**) Fruiting body produced on water agar baited with *E. coli* K-12 bait. (**g**) Optically refractile and rounded myxospores from a fruiting body produced on water agar. (**h**) Flexuous and\or slightly tapering vegetative rod cells obtained from CY–H broth. Petri dish diameter is 15 mm (**a**,**b**). Stereophotomicrograph (**c**–**f**). Phase-contrast photomicrograph (**g**,**h**).

**Figure 2 microorganisms-10-01262-f002:**
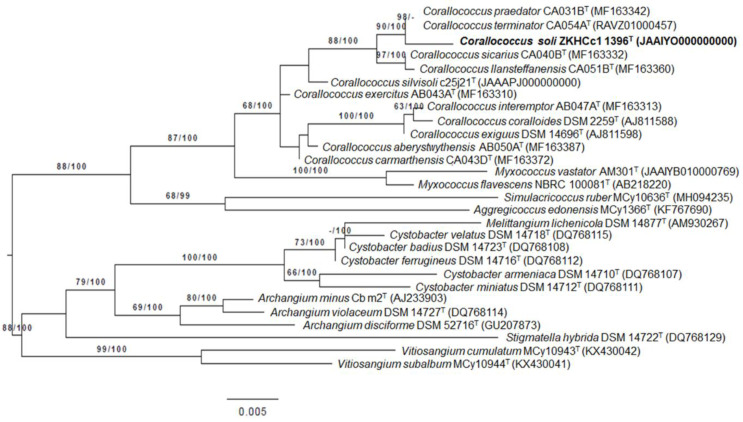
Phylogenetic tree based on 16S rRNA gene sequence of strain ZKHCc1 1396^T^ and related type strains. ML tree inferred under the GTR + GAMMA model and rooted by midpoint rooting. The branches are scaled in terms of the expected number of substitutions per site. The numbers above the branches are support values when larger than 60% from ML (left) and MP (right) bootstrapping. The ML bootstrapping did not converge; hence, 1000 replicates were conducted, and the average support was 70.64%. The MP bootstrapping average support was 84.24%.

**Figure 3 microorganisms-10-01262-f003:**
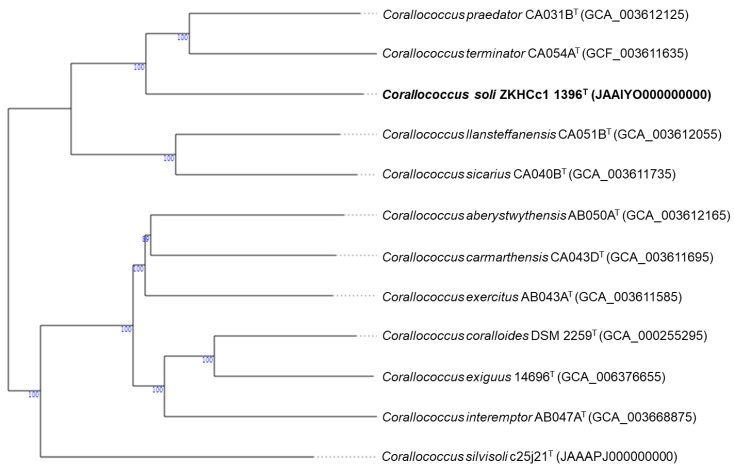
Genome BLAST Distance Phylogeny (GBDP) shows *Corallococcus soli* ZKHCc1 1396^T^ and the closely related *Corallococcus* type strain genomes curated in the genome server (TYGS) database. The numbers at the node represent > 60% GBDP pseudo-bootstrap confidence support (based on 100 replications and, on average, 98.8% branch support).

**Figure 4 microorganisms-10-01262-f004:**
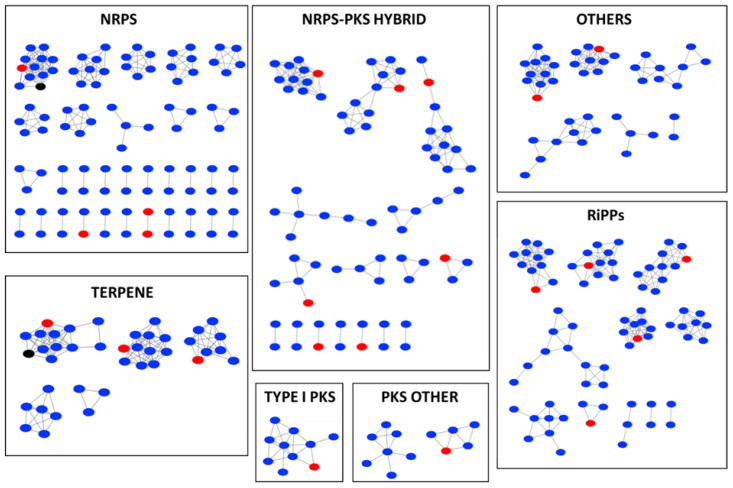
BiG-SCAPE BGC sequence similarity networks among myxobacteria in the genus *Corallococcus* and the related sequences from MIBIG database. Strain ZKHCc1 1396^T^ (red), MIBIG database related sequence (black), and all type strains of *Corallococcus* species (blue): *Corallococcus exercitus* AB043A^T^, *C. interemptor* AB047A^T^, *C. aberystwythensis* AB050A^T^, *C. praedator* CA031B^T^, *C. sicarius* CA040B^T^, *C. carmarthensis* CA043D^T^, *C. llansteffanensis* CA051B^T^, *C. terminator* CA054A^T^, *C. coralloides* DSM 2259^T^, *C. exiguus* DSM 14696 ^T^, and *C. silvisoli* c25j21^T^. Singletons were removed from the analysis.

**Figure 5 microorganisms-10-01262-f005:**
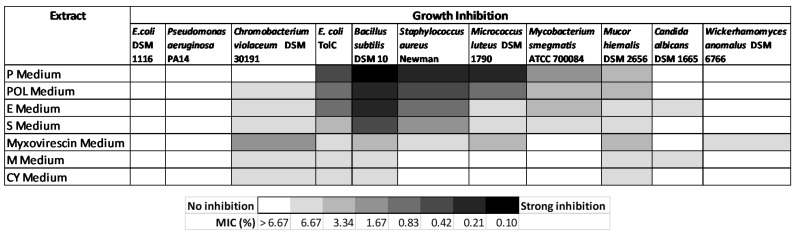
Heat map of antimicrobial activity of the extract of strain ZKHCc1 1396^T^.

**Figure 6 microorganisms-10-01262-f006:**
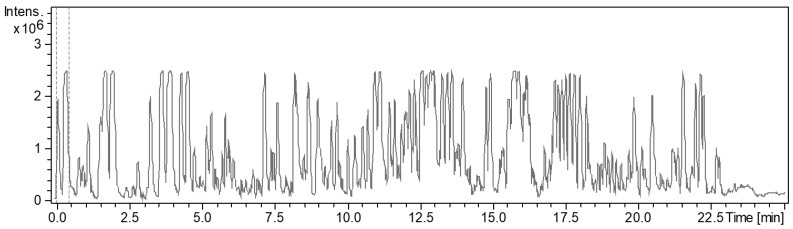
HPLC chromatogram of strain ZKHCc1 1396^T^ crude extract obtained after cultivation in P medium.

**Table 1 microorganisms-10-01262-t001:** Differentiating characteristics of strain ZKHCc1 1396^T^ and genomic information compared with type strains of *Corallococcus*.

Strains:	1	2	3	4	5	6	7	8	9	10	11	12
Temp. (°C): **30**	++	++	++	+	+	++	+++	++	++	++	+++	+++
**35**	+++	++	+	++	-	++	++	+	-	+	+	++
pH: 5	-	+	+	-	-	-	+	-	-	-	+	ND
**6**	+++	++	+	-	+	+	+	+	+	++	++	+++
**7**	+++	+++	++	+	+	++	+++	++	++	++	+++	+++
**8**	+++	++	++	-	-	++	+++	++	++	+++	+++	++
**9**	++	++	++	-	-	++	+++	++	++	+++	+	+
Biochemical												
**Esculin test**	-	-	+	-	+	+	+	+	+	+	-	+
**Gelatine test**	+	+	+	-	-	+	+	-	+	+	+	+
**Glucose assimilation**	-	+	-	-	-	-	-	-	-	-	-	+
**Maltose assimilation**	-	-	-	-	-	-	-	-	+	-	+	+
**Nitrate reduction**	-	-	+	-	-	-	-	-	-	-	-	-
**Antibiotic sensitivity**												
**Cefotaxime**	-	+	+	+	+	+	-	+	+	+	+	-
**Ceftazidime**	-	+	+	+	+	+	+	+	+	+	+	-
**Gentamicin**	+	+	+	+	+	+	-	+	+	+	+	-
**Trimethoprim-** **sulfamethoxazole**	+	-	-	-	-	-	-	-	-	-	-	-
Genome comparison:												
**Contigs**	68	961	459	625	1491	802	530	1244	863	1	36	62
**Genome size (Mb)**	9.44	10.15	9.47	9.98	10.51	10.39	10.79	10.53	10.35	10.08	10.41	9.23
**Mol% GC**	69.8	70.2	70.0	70.0	69.7	70.2	69.9	70.3	69.5	69.9	69.6	69.8
**No. of Gene**	7445	8611	7892	8353	9011	8442	8959	8867	8506	8148	8416	7412
**Pseudogene**	140	272	216	211	286	307	257	270	197	123	159	131
**No. of Protein**	7248	8271	7612	8079	8661	8072	8639	8539	8247	7952	8192	7221
**rRNA**	3	7	7	6	7	6	6	3	3	9	3	3
**tRNA**	50	57	53	53	54	52	53	51	55	60	58	53
**Other RNA**	4	4	4	4	3	5	4	4	4	4	4	4

1 Strain ZKHCc1 1396^T^ (Accession No. JAAIYO000000000), 2 *Corallococcus exercitus* AB043A^T^ (Accession No. RAVW00000000), 3 *C. interemptor* AB047A^T^ (Accession No. RAWM00000000), 4 *C. aberystwythensis* AB050A^T^ (Accession No. RAWK00000000), 5 *C. praedator* CA031B^T^ (Accession No. RAWI00000000), 6 *C. sicarius* CA040B^T^ (Accession No. RAWG00000000), 7 *C. carmarthensis* CA043D^T^ (Accession No. RAWE00000000), 8 *C. llansteffanensis* CA051B^T^ (Accession No. RAWB00000000), 9 *C. terminator* CA054A^T^ (Accession No. RAVZ00000000), 10 *C. coralloides* DSM 2259^T^ (Accession No. CP003389), 11 *C. exiguus* DSM 14696^T^ (Accession No. JAAAPK000000000), and 12 *C. silvisoli* c25j21^T^ (Accession No. JAAAPJ000000000). The genomes were compared using the NCBI Prokaryotic Genome. Growth indicators for pH and temperature: -, no growth; +, fair; ++, moderate; +++, best. Growth indicators for biochemical and antibiotic sensitivity tests: -, no growth; +, growth; ND, not determined. The data on pH, temperature, biochemical, antibiotic sensitivity for strains 2–12 were obtained from the previous studies of [22,63].

**Table 2 microorganisms-10-01262-t002:** Cellular fatty acid profile of strain ZKHCc1 1396^T^.

Fatty Acid	%
**C_10:0_**	tr
**C_14:0_**	tr
**C_14:1_**	tr
**C_15:0_**	tr
**C_16:0_**	1.3
**C_16:1_**	0.9
**C_18:0_**	1.3
**C_18:2_ ω6,9 all *cis***	0.2
**C_18:3_ω6,9,12 all *cis***	1.3
**C_16:0_ 2-OH**	0.1
**C_16:0_ 3-OH**	tr
Total SCFA**:**	5.1
***iso*-C_11:0_**	0.3
***iso*-C_12:0_**	tr
***iso*-C_13:0_**	2.4
***iso*-C_14:0_**	1.9
***iso*-C_15:0_**	15.8
***iso*-C_15:1_**	1.9
***iso*-C_16:0_**	5.6
***iso*-C_16:1_**	0.5
***iso*-C_17:0_**	9.4
***iso*-C_17:1_**	11.7
***iso*-C_17:2_**	0.9
***iso*-C_15:0_ 3-OH**	5.8
***iso*-C_16:0_ 2-OH**	0.7
***iso*-C_16:0_ 3-OH**	tr
***iso*-C_17:0_ 2-OH**	31.0
***iso*-C_18:0_ 2-OH**	0.1
***iso*-C_15:0_ OAG**	1.3
***iso*-C_16:0_ OAG**	tr
***iso*-C_15:0_ DMA**	5.0
Total BCFA**:**	94.2

tr, trace amount (below 0.1%).

**Table 3 microorganisms-10-01262-t003:** The similarity of strain ZKHCc1 1396^T^ to *Corallococcus* type strains based on ANI and dDDH.

	ANI/dDDH Value (%)											
dDDH\ANI	1	2	3	4	5	6	7	8	9	10	11	12
**1**	100	92	91	89	89	87	87	87	86	86	86	87
**2**	44	100	91	88	87	84	84	84	85	86	87	87
**3**	43	50	100	88	87	84	84	84	84	85	86	87
**4**	36	36	35	100	92	85	85	86	85	86	86	87
**5**	36	37	35	50	100	85	85	85	85	86	87	88
**6**	31	32	31	32	32	100	91	91	90	92	92	89
**7**	31	31	30	31	32	47	100	91	90	91	91	89
**8**	31	31	31	31	33	48	48	100	90	91	92	89
**9**	30	30	30	30	31	42	43	42	100	92	92	88
**10**	30	31	30	30	31	44	44	44	46	100	94	88
**11**	30	30	30	30	31	43	43	44	44	54	100	88
**12**	32	32	31	32	33	36	35	36	34	35	34	100

The strains and their genome accession number included in this analysis are the following: 1 Strain ZKHCc1 1396^T^ (JAAIYO000000000), 2 *C. praedator* CA031B^T^ (RAWI00000000), 3 *C. terminator* CA054A^T^ (RAVZ00000000), 4 *C. sicarius* CA040B^T^ (RAWG00000000), 5 *C. llansteffanensis* CA051B^T^ (RAWB00000000), 6 *C. exercitus* AB043A^T^ (RAVW00000000), 7 *C. aberystwythensis* AB050A^T^ (RAWK00000000), 8 *C. carmarthensis* CA043D^T^ (RAWE00000000), 9 *C. interemptor* AB047A^T^ (RAWM00000000), 10 *C. coralloides* DSM 2259^T^ (CP003389), 11 *C. exiguus* DSM 14696^T^ (JAAAPK000000000), 12 *C. silvisoli* c25j21^T^ (Accession No. JAAAPJ000000000). The ANI values are given above the diagonal grey area (values of 100%), while dDDH are shown below the diagonal grey area (values of 100%).

**Table 4 microorganisms-10-01262-t004:** Percentage similarity of the predicted biosynthetic gene cluster (BGC) of strain ZKHCc1 1396^T^ and *Corallococcus* type strains.

Corallococcus Species	Accession Number	Percentage Similarity of BGC											
		1	2	3	4	5	6	7	8	9	10	11	12
** *C.* ** ***soli* ZKHCc1 1396^T^**	JAAIYO000000000	100	100	83	100	100	88	100	100	100	-	-	-
** *C. praedator* ** **CA031B^T^**	RAWI01000000	100	80	33	100	100	-	-	-	100	-	-	-
** *C. terminator* ** **CA054A^T^**	RAVZ01000000	100	80	83	100	100	88	100	100	100	100	-	-
** *C. sicarius* ** **CA040B^T^**	RAWG01000000	45	80	75	100	-	22	-	-	100	-	-	-
** *C. llansteffanensis* ** **CA051B^T^**	RAWB01000000	27	80	83	100	100	-	-	100	100	-	100	-
** *C. exercitus* ** **AB043A^T^**	RAVW01000000	90	80	33	100	100	-	-	100	100	100	-	-
** *C. aberystwythensis* ** **AB050A^T^**	RAWK01000000	63	100	75	100	-	-	-	100	100	-	-	100
** *C. carmarthensis* ** **CA043D^T^**	RAWE01000000	100	100	83	100	100	-	-	-	100	-	-	-
** *C. interemptor* ** **AB047A^T^**	RAWM01000000	100	60	83	100	-	-	-	100	100	-	-	-
** *C. coralloides* ** **DSM 2259^T^**	CP003389	100	100	83	100	-	-	-	-	-	-	-	-
** *C. exiguus* ** **DSM 14696^T^**	JAAAPK010000000	100	100	83	100	-	-	100	-	-	-	-	-
*C. silvisoli* c25j21^T^	JAAAPJ000000000	100	100	83	100	100	-	-	100	100	100	-	-

1 Carotenoid, 2 VEPE/AEPE/TG-1, 3 myxochelin A/myxochelin B, 4 geosmin, 5 anabaenopeptin NZ857/nostamide A, 6 dawenol, 7 1-nonadecene/(14Z)-1, 14-nonadecadiene, 8 1-heptadecene, 9 rhizomide A/rhizomide B/rhizomide C, 10 icosalide A/icosalide B, 11 xenotetrapeptide, 12 bicornutin A1/Bicornutin A2. -, no similarity.

**Table 5 microorganisms-10-01262-t005:** Fraction analysis of strain ZKHCc1 1396^T^ crude extract obtained after cultivation and extraction in P medium.

Fraction	RT (min)	Major Ion		Number of Hits in DNP
		m/z	Ion	
1	1.87–3.99	211.1440 233.1257	[M + H]^+^ [M + Na]^+^	28
2	4.10–6.00	245.1282 267.1100	[M + H]^+^ [M + Na]^+^	61
3	6.09–8.02	209.1645	[M + H]^+^	21 (2 from *Chondromyces crocatus*)
4	8.12–10.03	223.1800	[M + H]^+^	7 (2 from *Chondromyces crocatus*)
		277.2156 295.2262 313.2368	[M-2H_2_O + H]^+^ [M-H_2_O + H]^+^ [M + H]^+^	59
		353.2292 683.4689	[M + Na]^+^ [2M + Na]^+^	552
5	10.18–11.99	277.2163 295.2264	[M-H_2_O + H]^+^ [M + H]^+^	135
		335.2192 647.4485	[M + Na]^+^ [2M + Na]^+^	11
6	12.10–14.01	319.2243	[M + H]^+^	1268
		317.2086 611.4280	[M + Na]^+^ [2M + Na]^+^	2
		279.2318 557.4566	[M + H]^+^ [M + Na]^+^	134
		277.2159 295.2255	[M-H_2_O + H]^+^ [M + H]^+^	135
		293.2086	[M + H]^+^	113
7	14.12–15.99	255.2317 277.2136 237.2210 509.4564	[M + H]^+^ [M + Na]^+^ [M-H_2_O + H]^+^ [2M + H]^+^	91
		325.2712	[M + H]^+^	169
8	16.11–18.01	353.2661 313.2736	[M + Na]^+^ [M-H_2_O + H]^+^	18
		257.2471 239.2365	[M + H]^+^ [M-H_2_O + H]^+^	40
		441.3549	[M + H]^+^	5
		485.3808	[M + H]^+^	1
		524.4517	[M + H]^+^	1
		529.4070	[M + H]^+^	3
9	18.11–19.98	686.4746 708.4566 1371.9401	[M + H]^+^ [M + Na]^+^ [2M + H]^+^	0
		589.4437 1155.8960	[M + Na]^+^ [2M + Na]^+^	3
		353.3021	[M + H]^+^	5

## Data Availability

Detailed data concerning this study is available upon request.

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
