# Peer review of "Corallococcus soli sp. Nov., a Soil Myxobacterium Isolated from Subtropical Climate, Chalus County, Iran, and Its Potential to Produce Secondary Metabolites"

_microorganisms, 2022, doi:10.3390/microorganisms10071262_

Round 1

Reviewer 1 Report

In the manuscript submitted by Babadi et al., the authors describe a novel myxobacterium and present preliminary data concerning its specialized metabolism. Originally isolated in 2017, the authors determine the taxonomic placement for Corallococcus soli from morphology, chemotaxonomy, and comparative genomics. This thorough evaluation supports the assignment as a novel species within the genus Corallococcus. Additional investigation of the biosynthetic gene clusters from C. soli and comparison with clusters from other members of Corallococcus provides some insight into the strain’s potential to produce novel metabolites. The authors also provide very preliminary metabolomic data as well as antimicrobial activities for crude extracts. Below are questions/comments for the authors as well as a few minor edits.

Comments/questions for the authors:

  1. According to recently proposed reclassification of most Deltaproteobacteria (Waite et al., Int. J. Syst. Evol. Microbiol. 2020;70:5972–6016), myxobacteria should instead be considered members of the phyla Myxococcota. The genus Corallococcus has been proposed to be in the updated family Myxococcaceae and not the suborder Cystobacterineae. This is just a comment for the authors and doesn’t require changes if they prefer to keep the classifications unchanged.

  1. Did the authors manually search all exact masses for each major ion using the dictionary or natural products website or was a more throughput method used?

  1. Analysis of ANI and dDDH data was done appropriately and is preferred over traditional 16SRNA analysis when comparing myxobacteria. Do the authors think that the highly fragmented genome data from C. soli (>60 contigs) as well as from many published genomes of Corallococcus type strains limit the accuracy of these techniques?

Minor edits:

  1. Inconsistent use of subscript in chemical formula throughout the text, tables, and SI should be corrected.

  1. The 1st paragraph on page 13 discussing production media and detected metabolites requires edits for clarity. For example, “The BPC chromatogram of P medium extract of strain ZKHCc1 1396T had more than twenty of high peaks in the range of 1.8-20 minutes.” What do the authors mean by “high peaks?” What is this relative to? I think that it’s generally clear that this is referring to detected ion intensity, but this should be made clearer.

  1. Figure 5 needs work. There are misspelled bacteria and inconsistent use of italics that need to be corrected. The color scheme of the heat map is not accessible to color blind readers. The use of percentage as the unit for MIC data doesn’t align with the experimental description. This should be clarified.

Reviewer 2 Report

In this study, the authors isolated and characterized a myxobacterial strain ZKHCc1 1396T and consequently propose it to be a novel species within the genus Corallococcus, which name is C. soli, sp. nov. They also report its potential to produce secondary metabolites based on the analysis of secondary metabolite-biosynthetic gene clusters (smBGCs) and antimicrobial activities of the culture extracts. Although the strain seems to be a new species as described here, the manuscript is unfortunately unacceptable for publication in this journal because the results on secondary metabolites are unclear and the part hardly seems to reach at the level for publication. Many parts are like a student report. Hence, my decision is ‘reject’. I strongly recommend the authors to withdraw the submission and reconsider the structure of the paper.

Major comments

  1. Abstract. Although this manuscript includes many results on secondary metabolites, they are not described in the abstract at all.
  2. Introduction. Aims of this study are not described in the introduction.
  3. Statement between Table 4 and Figure 4. It is hard for readers to understand the statement and what Figure 4 means.
  4. L11-12, p13. The statement is inappropriate because Table 5 shows many hits.
  5. Table 5 is hard for readers to understand.

Minor comments

1) L18-20, p2. The references 23 to 26 are inappropriate.

2) L9-10, section 2.1, p2. ‘E. coli K-12 DSM 498’ includes two strain numbers, K-12 and DSM 498.

3) L21, p3. acid study was performed -> acids were analyzed

4) L3, p5. Did the authors transfer all the resultant cultures (20 ml)? If not, indicate the volume.

5) L8, p5. Describe the volume of acetone used for extraction.

6) L11, p3. screened?

7) L16-25, p3. It does not seem significant to determine MICs for crude extracts.

8) Please locate Figures and Tables just after the paragraph where cited. The order and position are not appropriate in this manuscript.

9) L8-9, p6. stains Gram-negative -> Gram stain-negative

10) L12-14, p6. Negative API ZYM reactions (0) to α-, β- galactosidase, β-glucuronidase, α-, β- glucosidase, N-acetyl-β-glucosaminidase, α-manno-sidase, and α-fucosidase. -> API ZYM reactions were negative (0) to α-, β- galactosidase, β-glucuronidase, α-, β- glucosidase, N-acetyl-β-glucosaminidase, α-manno-sidase, and α-fucosidase.

11) L32-38, p6. Rewrite the part.

12) Table 1. Are ‘Contigs’, ‘No. of Gene’, ‘Pseudogene’ and ‘No. of Protein’ really differentiating characteristics as shown in the title?

13) Table 1. Biochemilcal?

14) Table 1. Why are the numbers of rRNA 7 in strains 2, 3 and 5? If they include two copies of rrn operon, they should be 6.

15) Footnote of Table 1. Table 1. -> 1)

16) Footnote of Table 1. Remove ‘Annotation Pipeline (PGAP).

17) Table 2. Fatty acids -> Fatty acid

18) Legend of Figure 2. Indicate the strain to make the root in the tree.

19) L13-14, p10. other Corallococcus type species -> type strains of other Corallococcus species

20) L17, p10. The bootstrap values do not support diverging but suggest that the strains are included in a clade.

21) Title of section 3.2. the Biosynthetic -> the Secondary Metabolite-Biosynthetic

22) L1-3, p13. P medium does not produce the extract.

23) L6, p13. Why can the author consider that the amounts are high?

24) L16, p15. the genome size -> the genome size of the type strain

Round 2

Reviewer 2 Report

The revised manuscript still includes many points that the authors should address and is hardly acceptable for publication.

L38-39

It is doubtful whether the analyses revealed the strain’s potential to produce novel compounds because most secondary metabolites, except for fraction 19, are not novel as suggested by the database searches.

L71-72

Unfortunately, the aim of this study is still unclear. The sentence shows not the aim but what the authors did in this study. I think the authors described the novel species and demonstrated the potential as approaches for some aims. Or, to describe and demonstrate them are the aims? If so, please state their scientific significance. 

L431-473

Rewrite the part. It is unclear what the authors like to report through the analysis. At least, the text should be edited by an expert. According to Table 4, the strain harbors 12 secondary metabolite-BGCs (smBGCs). In contrast, Figure 4 show it harbors 23. The network suggests that all the 23 smBGCs are not novel because their orthologs are present in other strains of this genus.

L484-486

Not convincing. ‘No hit’ in the text never means the product are novel because strains belonging to other genera may produce the compounds.

L487

As commented in parentheses by the authors, what are they? Indicate here. Please never submit an incomplete manuscript.

Figure 6

Many peaks must have been derived from components in the P-medium. Please compare chromatograms of extracts between the strain’s culture and P-medium and annotate only the peak of secondary metabolites produced by the strain.

Table 5

This table includes only 19 fractions, but the authors describe Fraction 19 in the text. Indicate each retention time.